

# Seismic Signal Classification of Snow Avalanches using Distributed Acoustic Sensing in Grasdalen, Western Norway

Franz Kleine[1], Charlotte Bruland[2], Andreas Wuestefeld[2], Volker Oye[2], and Martin Landrø[1]

[1]NTNU Norwegian University of Science and Technology, Trondheim, Norway
[2]NORSAR, Kjeller, Norway

**Correspondence:** Franz Kleine (franz.kleine@ntnu.no)





**Abstract.** We show the usage of Distributed Acoustic Sensing for analyzing seismic signals from snow avalanche events. For three winter seasons we continuously recorded seismic data using Distributed Acoustic Sensing (DAS) on a section of a standard telecommunication fiber along a mountain road in Grasdalen, western Norway. Multiple snow avalanche events were registered, alongside various other signals such as road traffic and detonations from remote avalanche triggering. We describe

5   signal characteristics of natural and manually triggered avalanche events and present a comparison with other observed signals in both time and frequency domain. Our frequency analysis shows that avalanche signals are most visible between 20 - 50 Hz. For larger avalanches, we observe weak low-frequency precursor signals, which correspond to the avalanche's approach. The more prominent high-amplitude signals appear to be produced by the snow masses impacting stopping cones or the steep terrain near the road. In one natural avalanche event, we interpret distinct spike signals as likely corresponding to the stopping snow

10   mass, based on similar findings from previous studies using geophones. Automatic detection, tested with simple STA/LTA thresholds in the 20 - 50 Hz range, presents challenges due to false positives from road traffic. Further refinement and testing are required to improve detection reliability in this complex environment. Our study represents an initial exploration into the application of Distributed Acoustic Sensing for snow avalanche detection, showcasing its potential as an effective monitoring tool for long road networks in mountainous regions.





# 1 Introduction

Snow avalanches and landslides pose a considerable risk for road safety and infrastructure in Norway (Bjordal and Larsen, 2009). According to Lunde and Njå (2019), this problem will become even more prevalent in the following years with the on-going climate change. Avalanche monitoring systems can help improve road safety by enabling immediate and automatic road closures when necessary. By gathering real-time observational data about avalanche activity, these systems can reduce the risk of vehicles entering avalanche prone areas. Additionally, assessing whether vehicles were present near the danger zone during an avalanche helps determine if a rescue mission is required. Hence, gathering observational data about avalanche activity is an important factor, optimally leading to a functioning monitoring and warning system.

Avalanche detection and monitoring have previously been done using either optical remote sensing or acoustic/seismic methods. Optical systems such as doppler-radar (Schreiber et al., 2001) or webcams (Fox et al., 2024) require direct line of sight and are limited in their spatial coverage. In the case of webcams the detection capability can be strongly decreased by bad weather conditions such as heavy snowfall. The second type of detection systems makes use of acoustic or seismic waves generated by avalanche events (Suriñach et al., 2001). They offer long detection distances up to a few kilometres (Marchetti et al., 2019) and immunity to bad weather conditions. Additionally, no direct line of sight is necessary, making it a reliable tool for automatic detection.

Seismic signals of snow avalanches were first recorded by St. Lawrence and Williams (1976). Suriñach et al. (2001) showed that seismic sensors can be used for studying avalanche characteristics. Analyzing seismic data from the same test site, Biescas et al. (2003) demonstrated that running frequency spectra show a gradual increase in higher frequency signals as the avalanche approaches the geophpone, resulting in a triangular shaped spectrum. Pérez-Guillén et al. (2016) studied frequency signatures of both dry and wet avalanches using geophones. For powder snow avalanches, the majority of seismic energy was found below 10 Hz, produced by the turbulent flow part at the front of the avalanche. For wet snow avalanches however, most of the seismic energy was generated by the dense flow regime, resulting in signals with a frequency content mainly above 10 Hz. Recording and localizing multiple avalanche signals can be achieved by utilizing a geophone array in combination with beam-forming algorithms (Lacroix et al., 2012) or statistical models (Heck et al., 2018).

Distributed Acoustic Sensing (DAS) is a technology that utilizes optical fibers for measuring dynamic strain or strain rate along the cable's axial direction. This is achieved by sending laser pulses into the fiber and recording the backscattering, subsequently extracting phase differences between different back-scattered light bits. The phase difference measurement can either be performed between the backscattering response from adjacent regions of fiber for the same incident pulse or between successive pulses but for the same region of fiber. Phase differences are then converted to strainrate or strain (Waagaard et al., 2021). DAS has been successfully tested and proven useful in a wide range of seismic applications, from oil and gas exploration (Liu et al., 2024) over monitoring of wells (Horst et al., 2015), urban activity (Lindsey et al., 2020), tides (Buisman et al., 2023), whales (Bouffaut et al., 2022) to microseismic events (Walter et al., 2020).

It offers both high spatial and temporal resolution, with sensing distances in the meter range (Waagaard et al., 2021) and a frequency resolution between 0.01 Hz (Taweesintananon et al., 2023) and 50 khZ (Parker et al., 2014), depending on the inter-



rogator. Since existing telecommunication fibers can be utilized there is a great potential for monitoring applications without

the immediate need of new installations. The key advantage over local installations like radar, geophones, and infrasound systems is that with a single interrogator, more than 100 km of fiber can be monitored continuously (Waagaard et al., 2021), potentially enabling the observation of multiple snow avalanche sites using just one setup.

First detection tests of triggered snow avalanches with DAS were conducted by Prokop et al. (2013). Using a fiber following an avalanche path, Fichtner et al. (2021) were able to observe snow avalanche signals and differentiate between flow regimes.

Paitz et al. (2023) employed an unsupervised clustering algorithm (Bayesian Gaussian Mixture Model) on a set of features extracted from DAS avalanche recordings from the same test site. Amongst other findings, they could recognize seismic far field signatures before the avalanche hit the fiber. This gives hope for a realistic monitoring system where the sensor position does not necessarily lie in the avalanche path but is instead situated some distance away from it, such as a telecommunication fiber in a valley. As mentioned, previous experiments were conducted on test sites with fibers following the avalanche paths.

With readily available telecommunication fibers along exposed roads there is a large potential of increasing avalanche monitoring systems without the need for further installations. Within this paper, we present results from a seismic avalanche test monitoring system in western Norway based on Distributed Acoustic Sensing (DAS), utilizing a standard telecommunication fiber along a mountain road as the receiver.



## 2 Monitoring Site Description

During the winter seasons of 2021/2022, 2022/2023 and 2023/2024 we recorded DAS data in the mountain valley of Grasdalen in the region of Innlandet, Norway. The measurement period was approximately 4 months for each season. The DAS survey was conducted on a 2 km section of a single mode standard telecommunication fiber buried next to the road.

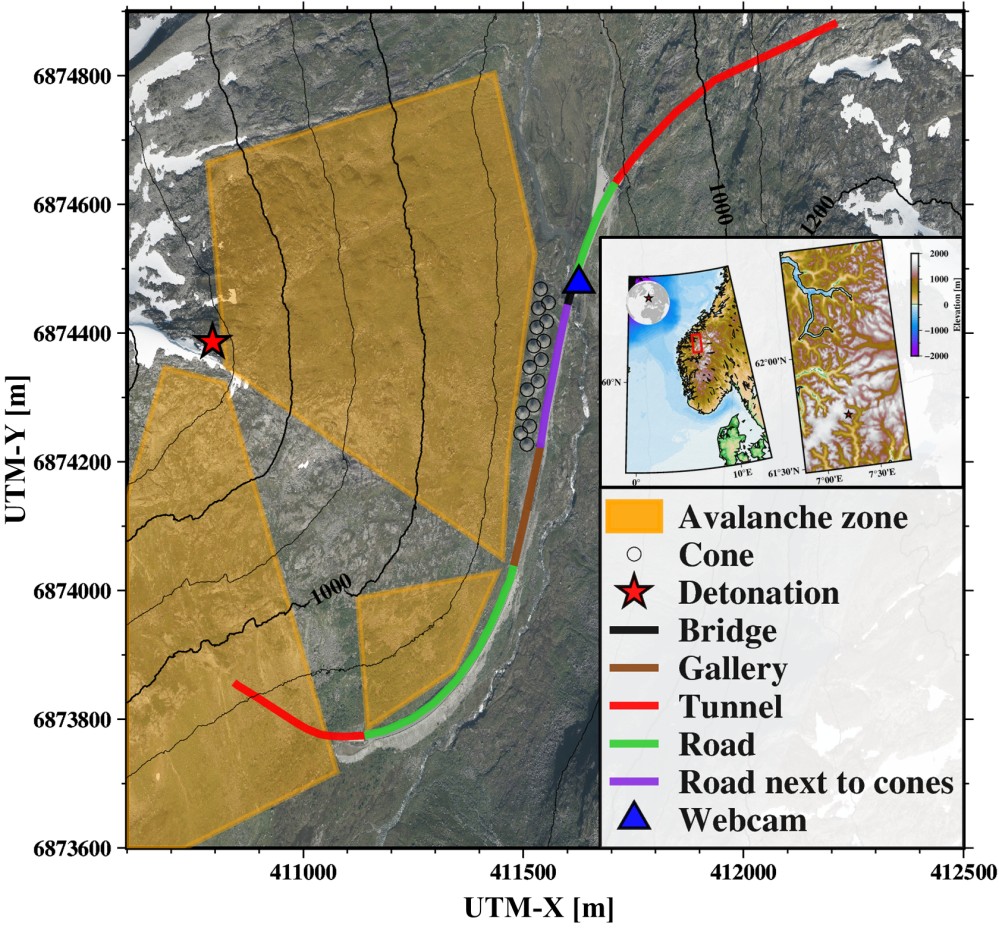

**Figure 1.** Overview of the Grasdalen Survey Setup. Note that the different sections of the fiber are color-coded based on the surroundings for better visual discriminability. The same color coding is used in alle subsequent figures. The altitude difference between neighbouring contour lines is 100 m. Aerial footage by Norwegian Mapping Authority (2020)

The environment of the road changes throughout the array: In the beginning and the end, the road is going into tunnels, whereas in the middle it is housed in a concrete gallery that shields it from rockfall and snow avalanches. On the part southwest of the

bridge, the road runs next to conical breaking mounds (labeled as cones) protecting it from avalanches. Due to the changing surroundings we expect to observe different noise levels throughout the array. The different cable environments are displayed in Figure 1 as colored lines. The mountainside is simultaneously monitored with a commercial, infrasound-based avalanche





detection system located 100 m east of the bridge. Additionally, a webcam located near the bridge was used for verification of avalanche activity and extent of avalanche runout zones.

The DAS array is monitoring two avalanche prone mountainsides: the east and the southeast face of Sætreskarsfjellet. The first one stretches from 1300 m.a.s.l. down to the valley floor at 875 m.a.s.l. Between 950 m.a.s.l. and 1250 m.a.s.l. the slope is mostly between 30 and 35 degrees steep, with rocky bands in between that are up to 45 degrees steep (NVE, 2024). Avalanches can be manually triggered by setting off explosives. The detonation point is located at the top of the mountainside at 1300 m.a.s.l. (see Figure 1)

A small section of the southeast face of Sætreskarsfjellet is also facing the open road south of the avalanche gallery. The altitude difference in this part is up to 175 m (between 1000 m.a.s.l. at the top and 835 m.a.s.l. next to the road) with a steepness of up to 45 degrees. The southeast mountain face above the southern tunnel is considerably higher and oftentimes steeper than 50 degrees based on (NVE, 2024).

Based on previous avalanche reports from Grasdalen (Varsom Regobs), the altitude difference and steepness, we expect
avalanches of up to size 3 (size level based on European Avalanche Warning Services (3/1/2024)) on the east as well as the southeast face. This means avalanches of a length over several 100 m with a total volume of 10000 $m^3$, capable of crossing the road and destroying cars. On the small stretch of open road below the southeast face avalanches of size of 2 can be expected.

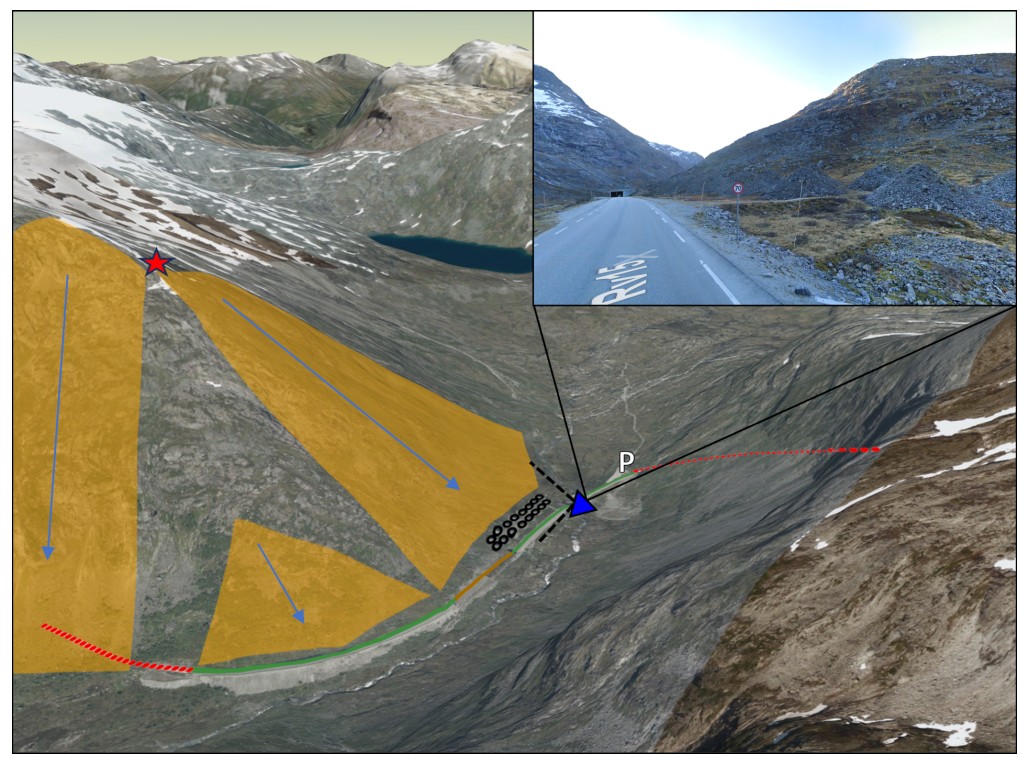

**Figure 2.** 3D view of the Grasdalen valley and the Survey setup (fiber sections are marked with same colors as in Figure 1). The inset image shows the east face of Sætreskarsfjellet as seen from the bridge. Avalanche prone areas area highlighted in orange, the detonation point is marked with a red star. The parking lot at start of the northeastern tunnel is marked with a 'P'. Aerial footage by Norwegian Mapping Authority (2020), inset image by © Google (2024).

An ASN OptoDAS interrogator was used for data acquisition. The acquisition parameters utilized for each season can be found in Table 1. For the last (2023/24) season, the sampling rate was reduced to 250 Hz. This adjustment was made because higher frequency data were not deemed necessary, since the dominant avalanche-related energy is found below 100 Hz (see below in the next section). Additionally this led to more manageable file sizes during data handling.

|                     | 2022 | 2023 | 2024 |
| ------------------- | ---- | ---- | ---- |
| channel spacing [m] | 2    | 2    | 2    |
| sampling rate [Hz]  | 500  | 500  | 250  |
| Gauge length [m]    | 3.1  | 5.1  | 5.1  |

**Table 1.** Acquisition parameters





# 3 Signal Classification

## 3.1 Types of Signals

During the surveying period the following types of signals were observed along the DAS array: Road traffic, detonations from avalanche triggering, both natural and artificially triggered snow avalanches, and background noise.

This section provides a comparison between these signal types, followed by an in-depth analysis of selected avalanche events. The avalanche events were identified in the DAS data using the avalanche catalog from Varsom Regobs (Varsom Regobs). This catalog is based on observations from webcams, an infrasound system in the valley and reports from human observers,

including personnel from the state road authority responsible for avalanche risk assessments.

An overview about the four most important recorded signal types in terms of number of appearance and importance is displayed in Figure 3. Figure 3a illustrates the signal from an artificially triggered avalanche on February 10, 2022. The avalanche impacted the cones and subsequently covered the road section between the bridge and the gallery with a half-meter thick layer of snow (Varsom Regobs, 2022-02-10). The signal is divided into two distinct areas: the initial smaller slide, starting at 10

seconds between 800 and 880 m (marked with 1 in Figure 3a), and the main slide (marked with 2) between 650 and 775 m. Figure 3b shows the signal of a naturally released snow avalanche from February 2, 2024. Three distinct features are observed: The first event between 7.5 s and 9 s (marked with 1) shows the highest amplitudes. Since those are observed west of the bridge near the cones, we interpret this as the slide hitting one or multiple of the cones. The second sharp event (2) occurs at the end of the open road section close to the tunnel. Since there is a steep incline from the riverbed up to a parking lot next

to the tunnel entrance, this signal likely corresponds to the snow mass impacting the ramp of the parking lot (marked with 'P' in Figure 2). The recurring activity in the area marked with 3 is interpreted as multiple smaller slides going off after the first one. Additionally, four short events (encircled in blue) with relatively high amplitudes, each lasting approximately 0.5 s, were detected in this area. Figure 3c shows signals generated by cars passing the valley from northeast to southwest on February 10, 2022. In total, 15 individual traffic sources can be identified. By comparing these signals with webcam-confirmed traffic

events, we conclude that the stronger amplitudes (marked with a 'T') likely correspond to trucks, while the other signals are attributed to passenger cars. For reference, a confirmed truck signal is added in the supplementary figures (A1). The strength of the individual traffic signals fluctuates across the array sections, with the gallery area appearing to be less sensitive. This variation is likely due to differences in the cable environment. The bridge section, in particular, exhibits noisy signals, which are further amplified when traffic passes over it. Similarly, the cattle grids at 10 m, 430 m, 705 m, and 1130 m show increased

signal noise under traffic conditions, as shown in Appendix A1. In Figure 3d, the explosion signal that triggered the avalanche shown in Figure 3a is displayed. It is visible throughout the whole array, with smaller amplitudes in the gallery section. The maximum duration is approximately 1 second. The first arrival appears as a slightly curved line with a maximum arrival time difference of 0.08 s between first and latest arrival for the whole array (not visible in the Figure due to the long time scale of 40 s). This is due to the varying distance between source and receivers as a simple direct wave modeling example shows

A2 a similar behaviour. To analyze the frequency content and distribution of the events, the seismic data were filtered into different frequency bands and compared using data from selected channels. For the artificial avalanche events, a channel was


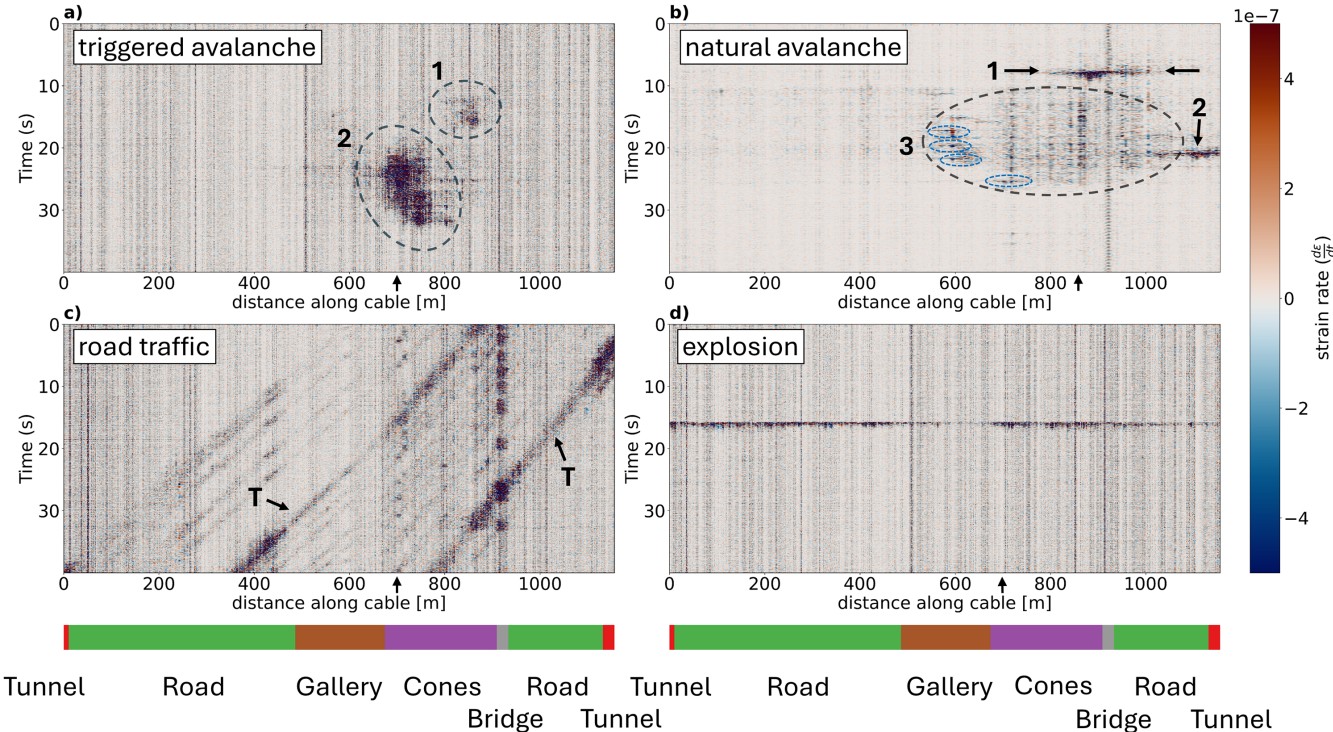

**Figure 3.** Strain-rate plots of four different signal types: a) Triggered snow avalanche from 10/02/2022 resulting in two signals that are marked with (1) and (2) b) natural snow avalanche from 02/02/2024 with three distinct features (marked with 1,2 and 3). Within the third event group, four noticeable spike events are encircled in blue. c) traffic signals from 10/02/2022 including two strong signals visible across all channels, associated with heavier vehicles like trucks (marked with 'T'). d) explosion signal from 10/02/2022 which triggered the avalanche shown in a).

A highpass filter (third order Butterworth) with a cutoff frequency of 0.5 Hz was used to suppress low frequency channel noise. Arrows mark the single channels that were used for frequency analysis (Figure 4).

selected in the most relevant section of the array adjacent to the cones, where the avalanche signal was strongest. This same channel was used to compare traffic and explosion data for consistency. Since the natural avalanche occurred closer to the bridge, a different channel was chosen where the signal for that event was more prominent. In Figures 4, the four different events from Figure 3 are compared, showing one channel each. Figure 4a represents an artificially triggered avalanche, Figure 4b a natural avalanche, Figure 4c road traffic, and Figure 4d an explosion used for avalanche triggering. Note that the y-axis scaling varies in Figure 4 to improve visibility and accommodate small values. For both the triggered and natural avalanches, most of the energy is found between 20 and 50 Hz. The triggered avalanche signal is visible across all frequency bands, with the strongest amplitudes between 20 and 50 Hz. The broad frequency response can likely be attributed to the size and extent of the event, as it impacted the cones and covered the road, including the area directly over the cable. The natural avalanche from February 2, 2024, shows a spike at the beginning, likely due to the avalanche hitting the cones. Signals from road traffic are





also prominently present in the 20–50 Hz range. Notably, the signal at 16 s is visible across all frequency bands. Individual car peaks are easily identifiable, especially in the 20-50 Hz band. The detonation signal is visible in the 20-100 Hz range. Since the snow avalanche signal are strongest in the 20-50 Hz band this frequency band was selected for later detection studies.

**Figure 4.** Frequency comparison of single channel (that are indicated with arrows in Figure 3) data for different event types in Winter: a) Triggered snow avalanche from 10/02/2022 b) natural snow avalanche from 02/02/2024 c) car signals from 10/02/2022 d) explosion signal from 10/02/2022. Note that y-axis ranges differ from subplot to subplot for better visibility.




**3.2   Signal evolution**

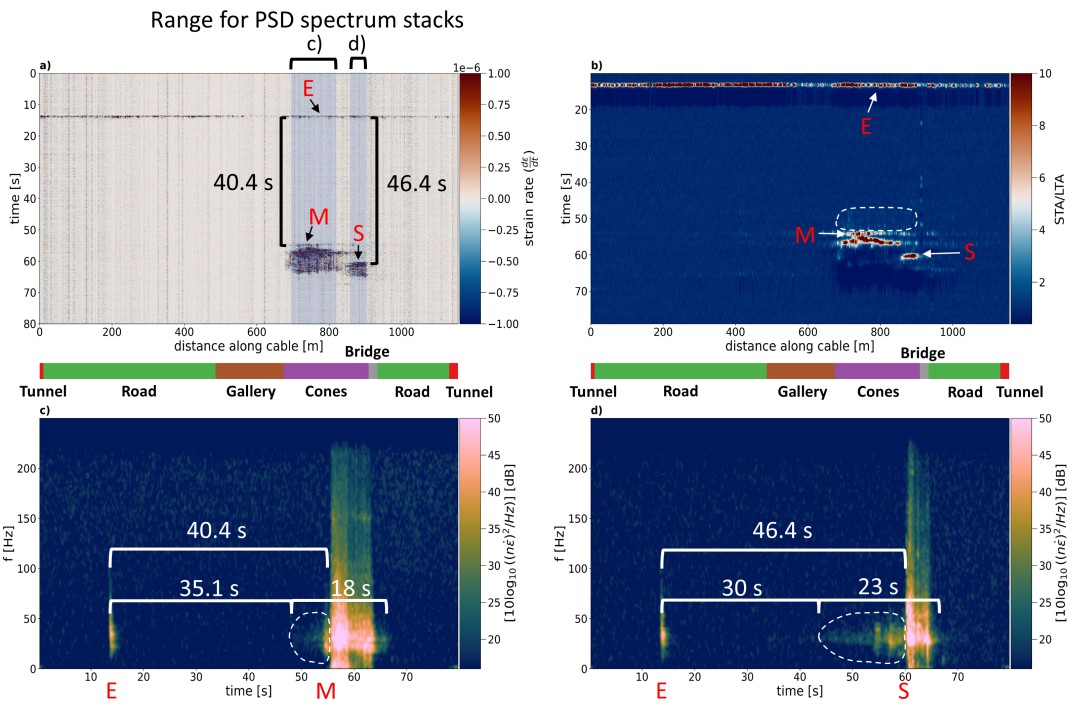

**Figure 5.** Temporal overview of the triggered avalanche from 10.02.2022.

a) Strain rate plot including explosion (E) and two avalanche signals (S for the small event and M for the main event). For reference, the time difference between the explosion and the two main signals is added (45.2 s for M, 36.4 s for S). Blue-shaded areas highlight the channels that are used to compute the stacked PSD spectrum plots shown in c) and d). A highpass filter (third order Butterworth) with a cutoff frequency of 0.5 Hz was used to suppress low frequency channel noise.

b) STA/LTA in 20-50 Hz band (short window: 1 s, long window: 10 s). Explosion (E) and two avalanche signals (S and M) are highlighted as in a). Precursor activity before the strong signal onsets is encircled in white. (c) and (d) Stacked PSD spectra for the two channel ranges indicated in (a). The time intervals between the explosion and the start of precursor activity, as well as between the explosion and the strong signal onsets (onsets are labeled with M and S, respectively), are shown with white brackets, along with the corresponding time differences. Precursor activity occurring before the strong signal onsets is encircled in white.

Natural Hazards
and Earth System
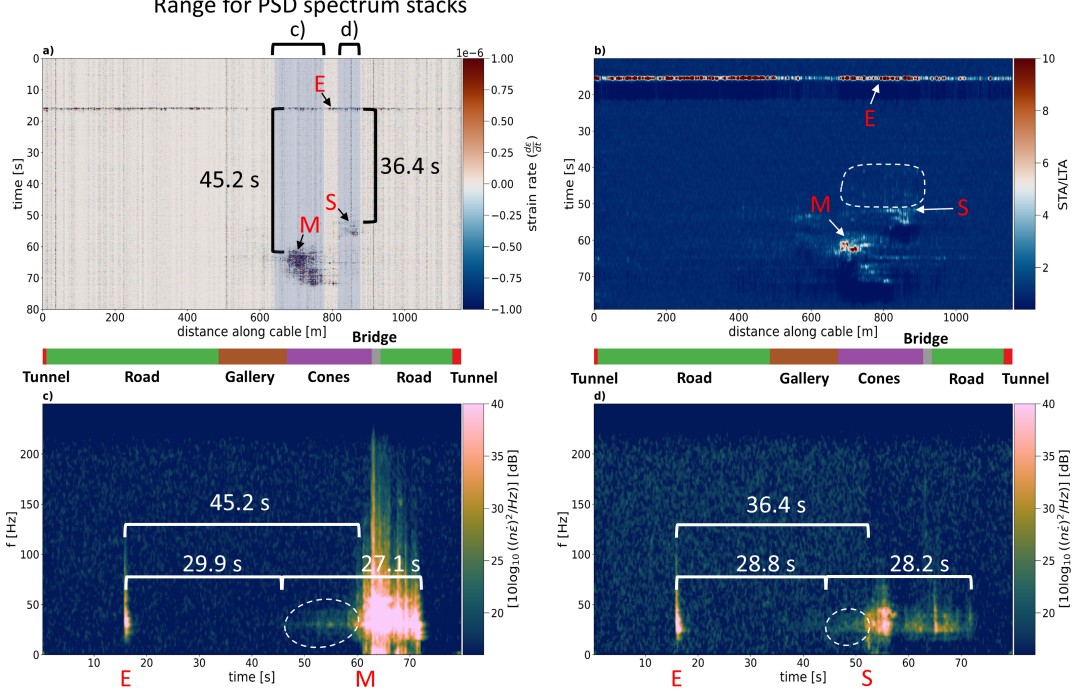

**Figure 6.** Temporal overview of the triggered avalanche from 10.04.2022.

a) Strain rate plot including explosion (E) and two avalanche signals (S for the small event and M for the main event). For reference, the time difference between the explosion and the two main signals is added (40.4 s for M, 46.4 s for S). Blue-shaded areas highlight the channels that are used to compute the stacked PSD spectrum plots shown in c) and d). A highpass filter (third order Butterworth) with a cutoff frequency of 0.5 Hz was used to suppress low frequency channel noise.

b) STA/LTA in 20-50 Hz band (short window: 1 s, long window: 10 s). Explosion (E) and two avalanche signals (S and M) are highlighted as in a). Precursor activity before the strong signal onsets is encircled in white. (c) and (d) Stacked PSD spectra for the two channel ranges indicated in (a). The time intervals between the explosion and the start of precursor activity, as well as between the explosion and the strong signal onsets (onsets are labeled with M and S, respectively), are shown with white brackets, along with the corresponding time differences. Precursor activity occurring before the strong signal onsets is encircled in white.

In this section, we present a more detailed analysis of the two largest triggered avalanches, focusing on their time evolution and signal characteristics. The analysis involves the computation of the Short Time Average over Long Time Average (STA/LTA) and Power Spectral Density (PSD). The STA/LTA was calculated using a frequency band of 20–50 Hz where the snow avalanche signal is the most prominent. We used a short window length of 1 second and a long window length of 10 seconds. In this case, the window lengths for the STA and LTA were chosen to effectively capture the onset of avalanche signals, including short, "spiky" events, such as those encircled in blue in Figure 3b. The power spectral density (PSD) spectra were calculated using a windowed Fast Fourier transform with Hanning windows of 0.5 s duration and 50% overlap. The PSD is stacked within the regions of avalanche activity and normalized by the number of stacked traces for comparison purposes. Figure 5 illustrates the time evolution of the avalanche triggered on February 10, 2022.





The strain rate plot (Figure 5a) shows the initial explosion at 16 seconds, followed by a small event occurring 36 seconds later, and the main event appearing 45 seconds after the explosion. While the strain-rate plot does not indicate significant activity before these events, minor disturbances are observed in the STA/LTA plot between 40 and 50 seconds (circled in white). In the stacked PSD spectra (Figures 5c and 5d, a weak signal between 20 Hz and 35 Hz emerges at 45.9 seconds for the first stacking window and at 44.8 seconds in the second. The onset of strong signals (M and S in Figure 5a) corresponds to a sudden

increase in power and a broadening of frequency content in the PSD spectra. For both stacking windows the activity stops at 73 seconds, yielding total event durations of 27.1 seconds and 28.2 seconds, respectively. This aligns well with the infrasound measurement of the avalanche, reporting a time of 28 seconds (Varsom Regobs, 2022-02-10). Given the size and impact of this avalanche, the infrasound measurement is considered reliable, consistent with previous findings on the effectiveness of infrasound in detecting large avalanches (Mayer et al., 2020).

The second analyzed event is a triggered avalanche from April 10, 2022, shown in Figure 6. The strain rate plot in Figure 6a reveals two distinct events: the main event (M) and a secondary, smaller event (S). Both events appear within a similar range of channels as those observed on February 10, 2022. This consistency suggests that the same two avalanche paths were active on both days. Notably, the time interval between the explosion and the onset of the strong signal is longer this time, measuring 40.4 seconds for the main event and 46.4 seconds for the smaller one. Due to the lack of additional data regarding the event,

the reason for this extended duration remains unknown. Although a clear precursor signal is not visible in Figure 6a, there is a slight indication of activity in the STA/LTA plot in Figure 6b. In the stacked PSD spectra, slight activity starts at 48.7 s for the main event (Figure 6c) and 43.6 s for the small event (Figure 6d). At the onset of the strong signals in the strain rate plot, we observe a corresponding increase in power in the PSD spectra, along with a broadening of the frequency content. This likely corresponds to seismic noise stemming from the snow mass hitting the cones. For both events, the activity in the PSD spectrum

resumes at 66.6 s. This leads to total event durations of 18 seconds and 23 seconds, respectively, which are significantly longer than the 6-second duration reported from the infrasound data. The signal evolution for the natural avalanche from February 2, 2024, is displayed in the appendix A3.



# 4 Discussion

We recorded seismic signals from snow avalanches using Distributed Acoustic Sensing (DAS) along a mountain road. The
avalanche signals varied significantly, depending primarily on the size of the avalanche and whether it reached the cones or the
road.

The highest amplitude signals, present over the full recordable frequency range between 0 and 250 Hz, were observed on
fiber stretches adjacent to the cones and the steep terrain incline leading up to the road. We believe that this could result from
the snow mass colliding with the mentioned features, producing high amplitude surface waves. We also detected pre-impact
signals using the Power Spectral Density Spectrum. Those lie in the frequency range from 20 - 50 Hz, likely caused by the
approaching avalanche (Figures 5 and 6).

For the natural avalanche on February 2, 2024, several short signals were observed at different parts of the cable as described
in Figure 3 (encircled in blue). Since three of those events are detected in the section of the avalanche gallery with no obvious
obstacles present, we associate them with several small parts of the slide coming to an abrupt stop. This would fit to the findings
of Suriñach et al. (2000) who reported short duration signals at the end of the avalanche signal trains, commonly referred to as
the stopping phase. They recorded artificially triggered snow avalanches using 3 - C geophones 1 - 3 km away from the
avalanche path. Similar signals were detected and analyzed in more detail by Suriñach et al. (2020), using geophones placed
along the avalanche path. In one case of their recorded events, a wet snow avalanche stopped before reaching the geophone,
presenting a scenario similar to the one observed in this study. The sudden stop of the snow mass produced a spike in the
seismic signal over the full frequency range of 1 - 40 Hz. Based on these findings, we are confident that the same explanation
is also valid for the aforementioned signals observed by our DAS system.

In the two triggered events, two spatially distinct signals suggest the presence of two slightly different paths for the avalanches.
Although this is not verified yet, we aim to investigate this further using avalanche simulations. Although we detect small
precursor signals for two large avalanches, they are weak and difficult to identify, making real-time detection challenging.
Additional instrumentation, such as extending the fiber up the mountainside, could improve this and also be used for studying
avalanche dynamics in more detail, as shown by Paitz et al. (2023).

Automated real-time avalanche detection remains challenging due to the variability of avalanche signals as well as the presence
of traffic noise. As an initial attempt, we tested STA/LTA threshold triggering in the 20-50 Hz frequency band with threshold
values between 5 and 10, focusing on the open road section next to the cones. While both triggered avalanches are picked
up by this method, smaller, less characteristic events such as the side events displayed in Figure A3a (labeled with 3) are not
detected. As mentioned, road traffic noise along the fiber path adds to the problem, since especially large trucks exhibit strong
signals present over the full frequency range. This effect is amplified at the cattle grids and the bridge where the noise spreads
over multiple channels as displayed in Figure A1a. This leads to numerous false positives, which could not be filtered out as
of now. Detecting smaller slides would require lowering the threshold further, making the system even more susceptible to
false positives from traffic noise. In conclusion, traffic noise needs to be identified and excluded through alternative methods
to improve detection accuracy.



One potential solution could involve using edge detection techniques to identify the typical diagonal lines that correspond to vehicles, followed by applying the Hough transform to extract these features. Given the large amount of available training data for traffic signals in this study, this approach could be further enhanced by integrating a Convolutional Neural Network (CNN) to improve line detection accuracy, as demonstrated by Xie et al.. The process would start with applying a conventional STA/LTA threshold trigger to a segment of data. If an event is detected, the resulting trigger matrix (a binary image containing only the trigger values) would be analyzed for diagonal (slanted) features, for example, using the Sobel operator. The Sobel-detected edges would then be processed with the Hough transform for line extraction. If no lines are detected, the event could be flagged as a potential avalanche.





## 5 Conclusion


We have made valuable observations of snow avalanches using Distributed Acoustic Sensing (DAS) along a mountain road. Identifying snow avalanches is critical, and DAS presents a promising method for utilizing existing infrastructure to address this challenge. Our results demonstrate the successful identification of both triggered and natural avalanches within the DAS data. We have characterized the observed signals, including those from road traffic and explosions, noting that avalanche signals
are most prominent in the 20–50 Hz frequency band. Weak precursor signals were observed for larger events, though these are insufficient for reliable automatic detection. We believe that the large amplitudes are generated when the snow mass collides with the snow-stopping cones next to the road and the slope leading up to the road near the northeastern tunnel. For natural avalanches, we observed short, spike-like signals across several channels, which we attribute to the snow mass coming to a stop. However, road traffic, particularly large trucks, generates high-amplitude signals across the entire frequency spectrum,
complicating efforts to develop automatic detection methods. Further work is necessary to enhance our ability to automatically detect these events, with the goal of ultimately achieving reliable automatic detection.




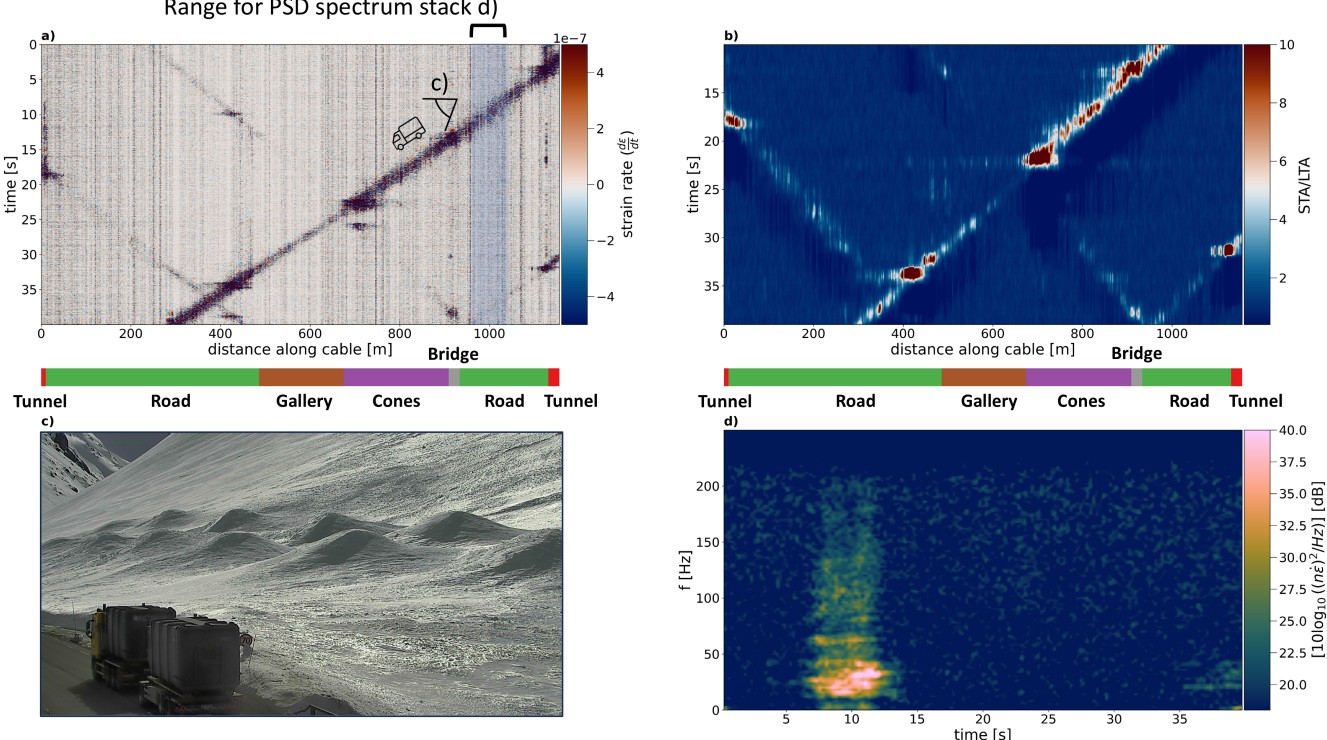

**Figure A1.** Road traffic signals from 16.04.2022. The strongest signal belongs to a two-axle truck passing the array from northeast to southwest.

a) Strain rate plot with signals of 6 individual vehicles. The strongest signal train is caused by a truck, passing the road from northeast to southwest. This was verified by the webcam footage shown in c). Webcam and truck position from the webcam image are marked. Note the noisy signals created by the vehicles passing over the kettle grids at 10 m, 430 m, 705 m and 1130 m as well as the bridge. The blue-shaded area highlights the channels that are used for the stacked PSD spectrum plot shown in d). A highpass filter (third order Butterworth) with a cutoff frequency of 0.5 Hz was used to suppress low frequency channel noise.

b) STA/LTA in 20-50 Hz band (short window: 1 s, long window: 10 s)

c) Webcam Image showing the truck next to the cones. Webcam and truck position are highlighted in a)

d) Stacked PSD Spectra for the two range of channels.





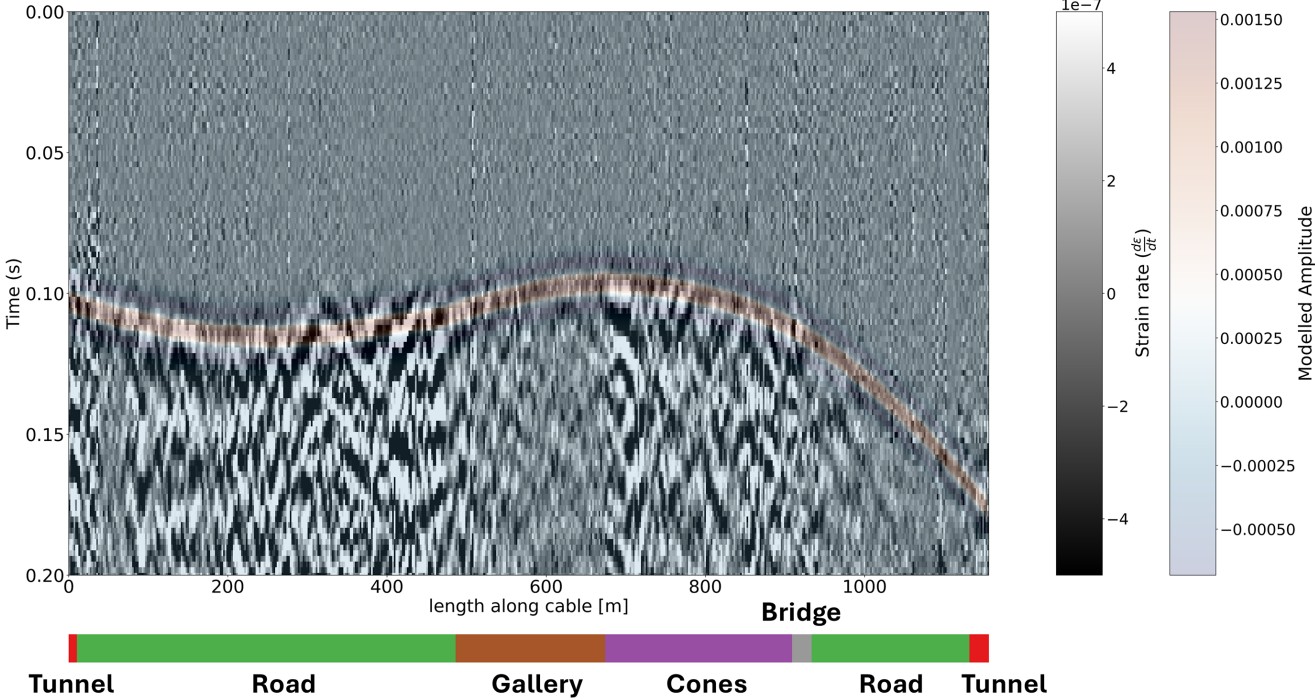

**Figure A2.** First Arrival of Explosion signal from 10.02.2022 - comparison of recorded and modeled signal.

Recorded strain rate plot is displayed in grayscale. A highpass filter (third order Butterworth) with a cutoff frequency of 0.5 Hz was used to suppress low frequency channel noise. The modelled first arrival (direct travel path, constant velocity of 2500 $\frac{m}{s}$, topography neglected) is overlayed.



**Figure A3.** Temporal overview of the natural avalanche from 02.02.2024.

a) Strain rate plot including two strong avalanche signals (labeled 1 and 2) and several less strong avalanche signals (labeled with 3). Four noticeable short events within this range are encircled in blue. Blue-shaded areas highlight the channels that are used to compute the stacked PSD spectrum plots shown in c) and d). A highpass filter (third order Butterworth) with a cutoff frequency of 0.5 Hz was used to suppress low frequency channel noise.

b) STA/LTA in 20-50 Hz band (short window: 1 s, long window: 10 s). The two strong avalanche signals (1 and 2) are highlighted as in a). The short events mentioned in a) are encircled in blue.

c) and d) Stacked PSD spectra for the two channel ranges indicated in (a). The total event time is indicated with white brackets. Precursor activity occurring before the strong signal onsets is encircled in white.

*Author contributions.* AW and VO initiated the research; AW gathered the data; FK analysed the data, plotted graphs and maps and wrote the manuscript; AW and CB reviewed graphs and assisted with data analysis; VO and ML reviewed and edited the manuscript.



*Data availability.* While we cannot upload the entire dataset due to its large size, we are preparing relevant data snippets and plan to upload
them to an online repository, which will be publicly available.

*Competing interests.* The authors declare that they have no conflict of interest.

*Acknowledgements.* This work was partially supported by the SFI Centre for Geophysical Forecasting under grant 309960. We thank all
participants in the DAS measurement campaign and result discussions. Special thanks to Statens Vegvesen for providing webcam footage,
avalanche and traffic data, and for giving us a general overview of the site. The colormaps used in this work were developed by Crameri
235 (2023).



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
