# Peer review of "Seismic Signal Characterization of Snow Avalanches using Distributed Acoustic Sensing in Grasdalen, Western Norway"

_Natural Hazards and Earth System Sciences, 2024_

## Referee Comment (RC1)

The present manuscript of Kleine et al. describes selected signals from avalanches recorded with the Distributed Acoustic Sensing (DAS) technology. The scientific question that the manuscript asks, is if DAS along telecommunication fibers paralleling mountain roads can be facilitated for avalanche detection and possible early warning to traffic related avalanche accidents. The presented dataset (3 full winter seasons) is to my knowledge to date one of the most comprehensive DAS data sets related to avalanche studies.

Overall, the manuscript is well written and should be accessible for the readership of NHESS that are not necessarily familiar with DAS. The used methods are standard and solid. The results are reasonable. Figures, language, and manuscript structure are in good shape. Concerning references, I personally would avoid citing conference abstracts, as they are not peer-reviewed.

What I am not so sure about, is to what extent the manuscript represents a substantial contribution to the understanding of natural hazards and their consequences. And here comes the crux. The present study is in many ways similar to Edme, P., Paitz, P., Walter, F., van Herwijnen, A., & Fichtner, A. (2023). Fiber-optic detection of snow avalanches using telecommunication infrastructure. *arXiv preprint arXiv:2302.12649*. However, that study is also still in the pre-print stage and therefore it is difficult for me to assess the novelty of the present manuscript. The significance of the present study lies in my opinion in the temporal extent of the data set and the scientific possibilities that come with it. The largest weakness of this study is in my opinion that this potential of the data is not being exploited, as only avalanche examples are presented and a more quantitative analysis is lacking.

In principle I recommend the present manuscript for publication in NHESS because 1) a to date novel data set is presented and 2) the avalanche observations are to my knowledge either first or second of its kind. However, in my opinion the potential of this study and especially of the data is much higher than it is presented currently. Therefore, I make some suggestions to the authors that would in my view substantially increase the comprehensibility to the reader as well as the relevance of this study.

I hope my review is helpful to the authors. I would be happy to discuss and respond to questions.

Best regards,
Dominik Gräff

**General Comments:**

*Novelty:* I would recommend making clearer what the novelty of the present manuscript is. The manuscript would benefit from emphasizing the main message. 'What do we learn?' was a question that I was asking myself throughout the manuscript.

*Language:* To me, the manuscript seems partially wordy and very descriptive (probably as my review. Sorry for that.). Whereas the descriptiveness is not necessarily bad, it is at times

difficult to filter out the relevant information and what the actual point of all the description is. Therefore, the manuscript could be more precise and concise. A few examples:
L131/132: 'Note that the y-axis scaling varies in Figure 4 to improve visibility and accommodate small values.' You can remove the entire sentence after 'Figure 4' without losing information. You save 50% of space, remove many unnecessary words, and therefore strengthen the important information.
Second example L154/155: 'The onset of strong signals (M and S in Figure 5a) corresponds to a sudden increase in power …' What strong signal in a PSD does not correspond to a strong power? The important part comes later in the sentence, namely that the frequency content is broadening.
Another example: L141-143: 'In this section, we present a more detailed analysis … and Power Spectral Density (PSD).' These two sentences are obsolete, because you repeat the exact same later. What would strengthen the section, would be to put the main finding up front in one sentence, or a sentence what the goal of the section is. This helps the reader to understand why you are doing the PSDs and the STA/LTA analysis.

Avoid using quantifying language without being precise. E.g. L192: 'slightly different paths'. How slightly different? 1m, 10m 100m? There is literally no meaning in the word 'slightly' here. Be explicit: '…different paths by about 100m.'

*Formatting:* I highly recommend using ISO 8601 for the date and time format. For readability in the text, a form like '1 January 1970 00:00' is usually chosen. Also, please also always include the time-zone or indicate '(UTC)' after your time stamp. I could find four different formatting types for date and time throughout the manuscript. Please be concise.

*Scientific Potential:*
The present DAS data set has certainly a large scientific potential for comparing various avalanches, and possibly avalanche types, over multiple seasons and years. I appreciate the focus of this paper on the avalanche signal content. However, in my opinion the large benefit of DAS, namely the spatial resolution over large apertures, comes short in the manuscript. The avalanche frequency content spatially resolved may allow for further insights how the avalanche propagates. Beamforming might be used to track the avalanche path down the slope. As of now, waveforms are shown in Fig. 4, however, their interpretation is limited to the signals' frequency content. Including a quantitative analysis, the 'results' section could be much stronger.

**Specific Comments/Questions:**
- Where is the interrogator located?
- What you are referring to as 'PSDs' in your manuscript, I would call 'spectrograms'. Usually, such spectrograms show the Power Spectral Density if not noted differently on the color bar, such as: Amplitude Spectral Density.
- Frequency filtering in spectrograms does not make sense. You can simply cut off frequencies that are not of interest or saturate the color map.
- What strain-rate signals do you actually measure with your DAS setup. I presume that from the explosion you record Rayleigh waves. Do you also expect to see Rayleigh waves for the signal of the cone-impacts? What do you measure when the avalanche propagates across the road. You shortly mention the seismic (presumably

Rayleigh) waves from the avalanche propagation down the slope. I think this would be worth to elaborate more on, since this signal would be the relevant one for early warning.

- I do not understand why the STA/LTA is so prominent in the manuscript. Yes, it is a standard detection method, but in your case, it seems to me that it does not perform any better than simply thresholding the absolute strain-rate data.
- The diverging colormap for STA/LTA implies a triggering threshold at the center. Why not use a monochromatic color map, or center your colormap around your trigger threshold?
- I don't understand why the manuscript elaborates so extensively about the duration of the events, or signals. What do the different timings mean? Avalanche propagation; propagation over the cones; can information about the avalanche speed be derived from these timings? Can that be linked to snow conditions, size, path? That would tremendously increase the significance of the manuscript.

**Line-specific Comments/Questions:**

**Abstract:**
L7: '… avalanches approach towards the fiber.'

**1 Introduction:**
L48: Typo: kHz instead of khZ
L59-61: Repetitive. I would recommend shortening or remove entirely.

**2 Monitoring Site Description**
L75ff: I get a bit lost reading all the slope aspects and must go forward and backward to Fig. 1 and within the paragraph. For me it would help labeling the slopes A,B,C or similar, if this is applicable. However, it looks like the paper only deals with avalanches from the easternmost slope. If so, is there a necessity to indicate the other slopes at all?

L92: Wordy. I would suggest: 'We recorded strain rate with an ANS OptoDAS interrogator with 2m channel spacing, variable gauge lengths between 3.1m and 5.1m and sampling rates of 250 and 500 Hz for different seasons.' Or something similar.

**3 Signal Classification**
**3.1 Types of Signals**

L101: Something is wrong with the wording of this sentence. Also, what makes a signal 'important'? Are these 4 signals most prominent or clean or representative?

L102ff: The timing information is missing. It would be great for reproducibility if the exact timing information in UTC is provided.

120: So why is the explosive signal not visible in Fig. 3a? What is the timing between the explosive signal and the avalanche signal. Are different frequency filters applied to a) and d)? - Note: This becomes clear later in the manuscript. However, it may be worth mentioning here.

L125: Some wording seems odd here.

L134/135: I don't understand that reasoning. A broad frequency content comes from a short signal. That fits well into your theory of an impulsive impact at the cones.

**3.2 Signal evolution**
L148: What is being stacked for the PSDs? The waveforms before calculating the PSDs? Or the PSDs of individual channels? In this case, the latter would be the appropriate way to avoid negative interference. Also, what you call PSDs, I would call 'spectrograms'. Of course, each spectrogram time segment represents the PSD.

L165: 'Due to the lack of additional data regarding the event, the reason for this extended duration remains unknown.' To me it's obvious that either one avalanche propagated faster than the other, or that the propagated distance to the cones differs. Is that a wrong or unreasonable assumption?

L168ff: This is important and should be described more concise. I start to understand only at this point, what you measure, as here you describe that you sense the seismic wave arrivals. It is written a bit unclear, such that I thought the 'seismic noise' comes from the impact on the cones, which does not make sense, because of the timing. Make clear that the 'seismic noise' corresponds to the avalanche propagation before the avalanche front hits the cones. Also, I think you should avoid the term 'seismic noise'. It is a clear signal. Probably also with coherent arrivals along DAS channels. I would assume that this signal is comprised from Rayleigh waves.

**4 Discussion**

L175: Where is this shown? You don't show plots where it gets clear if the road is reached or not it is not clear to me what signal content corresponds to what size of avalanche and how it varies significantly.

L180: This sounds correct to me. It's much clearer than what is written in L168ff.

L192: Remove 'slightly'.

L193: 'Although this is not verified yet, we aim to investigate this further using avalanche simulations.' Where? If you don't present it, there is no point in writing this.

L216: 'We have made valuable observations...' I would suggest letting the reader decide if the observations are valuable.

L224: 'across the entire frequency spectrum' This should be more specific.

**Figures:**

General: The unit on any axis representing strain rate should be s$^{-1}$, not $d\epsilon/dt$, as $d\epsilon/dt$ is not a unit.

Fig. 1: I got confused by the zoomed inset map (rightmost). I'd recommend a map that also includes roads, in particular the road that represents your fiber route. I thought your study site is between Geiranger and Erdal. The star in that zoomed inset is south-east of Jostedalsbreen.

Fig. 2: I personally don't like having a 2d birds-eye view and a 3d view of the study region in the same manuscript, because it's repetitive. I'd say in this case it's ok. However, I personally would put a figure like this in the supplementary information, as it is not essential for the manuscript.

Fig. 3: I strongly recommend using ISO 8601 as an international standard of displaying date and time. In your notation implies the American format and will certainly lead to misunderstandings.

Fig. 4: I start wondering why you always show the 40s-time window for the explosion. I understand that you do it for consistency in Fig. 3. But this figure is really about the waveforms, correct? If the figure is only about the power per frequency interval, it would be much nicer to show a 'spatial spectrogram' with spectral power plotted versus DAS channels.

Fig. 5: It is unclear to me, if the for the stacked PSDs the DAS channel waveforms were stacked, or if the PSDs of individual DAS channels were stacked.

The upper row (a,b) has the time on the y-axis, whereas the lower row (c,d) has time on the x-axis. To me that's inconsistent, but not wrong. So it is your choice, but for future publications I recommend being consistent. For me, it helps understanding such multi-panel figures.

Figure label: A new date format again. I recommend using ISO 8601.

Fig. 6: Similar to Fig. 5

Figure A1: a) I see 4 diagonal lines. Therefore 4 vehicles, not 6. Am I overseeing something?

Figure A2: This is a great figure. We learn so much from it. There are coherent phase arrivals, you can fit a seismic wave speed (presumably a Rayleigh wave). As no information about the modeling is provided, I do not know what the amplitudes refer to. They also do not have a unit. In my opinion it is already enough to only have a constant velocity-travel time curve, i.e. a curved line representing the first arrival. Due to the steep topography, I think you should do it in 3D. This should result a longer distance traveled and therefore higher wave speed. I would expect a value between 2500-3300 m/s for Rayleigh waves.

**Tables:**

Table 1: I think this could go in the supplementary information as it is not essential to understand the manuscript.

**Data availability.**
This seems useless. In my opinion it is ok if the data is not publicly available. I know that this does not agree with some journal policies. I face similar problems as you do with my DAS data. I would simply write: 'At the moment, the DAS data of this study not publicly available. Access can be granted on individual request.'

---

## Author Comment (AC1)

**Answer to Dominik Gräff**

**Thank you for your detailed comments and suggestions.**

**General Comments:**

*Novelty:* I would recommend making clearer what the novelty of the present manuscript is. The manuscript would benefit from emphasizing the main message. 'What do we learn?' was a question that I was asking myself throughout the manuscript.

**Three points should be mentioned:**

- **Real-World Application: We use a fiber-optic cable that was not deployed for the purpose of avalanche monitoring. This case study highlights both the potential and the limitations of utilizing existing telecommunication infrastructure.**
- **Signal Analysis: We compare avalanche signals to other sources, such as traffic, to point out the challenges inherent in detecting avalanches under realistic conditions.**
- **Extended Monitoring Period: Our dataset spans a longer time frame than previous studies.**

**Apart from the mentioned preprint (Edme et al., 2023), to our knowledge no other study has examined avalanche signals in a similar setup (using a cable setup that was not specifically laid out for monitoring purpose).**

*Language:* To me, the manuscript seems partially wordy and very descriptive (probably as my review. Sorry for that.). Whereas the descriptiveness is not necessarily bad, it is at times difficult to filter out the relevant information and what the actual point of all the description is. Therefore, the manuscript could be more precise and concise.

A few examples:

L131/132: 'Note that the y-axis scaling varies in Figure 4 to improve visibility and accommodate small values.' You can remove the entire sentence after 'Figure 4' without losing information. You save 50% of space, remove many unnecessary words, and therefore strengthen the important information. Second example L154/155: 'The onset of strong signals (M and S in Figure 5a) corresponds to a sudden increase in power …' What strong signal in a PSD does not correspond to a strong power? The important part comes later in the sentence, namely that the frequency content is broadening. Another example: L141-143: 'In this section, we present a more detailed analysis … and Power Spectral Density (PSD).' These two sentences are obsolete, because you repeat the exact same later. What would strengthen the section, would be to put the main finding up front in one sentence, or a sentence what the goal of the section is. This helps the reader to understand why you are doing the PSDs and the STA/LTA analysis. Avoid using quantifying language without being precise. E.g. L192: 'slightly different paths'. How slightly different? 1m, 10m 100m? There is literally no meaning in the word 'slightly' here. Be explicit: '…different paths by about 100m.'

**We use more concise language in the revised manuscript and specifically address the mentioned examples.**

*Formatting:* I highly recommend using ISO 8601 for the date and time format. For readability in the text, a form like '1 January 1970 00:00' is usually chosen. Also, please also always include the time-zone or indicate '(UTC)' after your time stamp. I could find four different formatting types for date and time throughout the manuscript. Please be concise.

**We adjusted the formatting type to ISO 8601.**

*Scientific Potential:*

The present DAS data set has certainly a large scientific potential for comparing various avalanches, and possibly avalanche types, over multiple seasons and years. I appreciate the focus of this paper on the avalanche signal content. However, in my opinion the large benefit of DAS, namely the spatial resolution over large apertures, comes short in the manuscript. The avalanche frequency content spatially resolved may allow for further insights how the avalanche propagates. Beamforming might be used to track the avalanche path down the slope. As of now, waveforms are shown in Fig. 4, however, their interpretation is limited to the signals' frequency content. Including a quantitative analysis, the 'results' section could be much stronger.

**Specific Comments/Questions:**

- Where is the interrogator located?

**The interrogator is located at a service point inside the Grasdalen tunnel (southern tunnel, see revised map).**

- What you are referring to as 'PSDs' in your manuscript, I would call 'spectrograms'. Usually, such spectrograms show the Power Spectral Density if not noted differently on the color bar, such as: Amplitude Spectral Density.

**Yes, with PSD we are referring to Power Spectral Density. This was adjusted to 'spectrogram' in the revised manuscript.**

- Frequency filtering in spectrograms does not make sense. You can simply cut off frequencies that are not of interest or saturate the color map.

**We assume that you refer to the highpass-filter that was applied at the start of processing (passing 0.5 Hz and above)? The difference between completely unfiltered data and the above mentioned highpass-filtered data in the spectrogram is negligible, only making the low frequency noise below 0.5 Hz visible.**

- What strain-rate signals do you actually measure with your DAS setup. I presume that from the explosion you record Rayleigh waves. Do you also expect to see Rayleigh waves for the signal of the cone-impacts? What do you measure when the avalanche propagates across the road. You shortly mention the seismic (presumably Rayleigh) waves from the avalanche propagation down the slope. I think this would be worth to elaborate more on, since this signal would be the relevant one for early warning.

**We think the Rayleigh wave is the dominating wave. The first arrival signals stemming from the detonation combined with the modelled first arrival travel time fits to an approximate velocity around 2.5 km/s, which fits to the rock type in the area (orthogneiss). However, this is difficult to prove for the case of avalanche signals. The cable is running perpendicular to the direction of the snow avalanches.**

- I do not understand why the STA/LTA is so prominent in the manuscript. Yes, it is a standard detection method, but in your case, it seems to me that it does not perform any better than simply thresholding the absolute strain-rate data.

**We found that STA/LTA gives a simpler trigger signal, corresponding to the onset of signals. While simple threshold triggering also detects the avalanche events, it also catches short impulsive signals (such as the small impulsive signals for the natural avalanche in Appendix 4 (a)). However, we did not mention that in the manuscript and included an overview about three different detection methods (simple threshold, STA/LTA and Kurtosis, see Attachment 4).**

- The diverging colormap for STA/LTA implies a triggering threshold at the center. Why not use a monochromatic color map, or center your colormap around your trigger threshold?

**This is a good point; we will adjust to a monochromatic color map in the updated manuscript.**

- I don't understand why the manuscript elaborates so extensively about the duration of the events, or signals. What do the different timings mean? Avalanche propagation; propagation over the cones; can information about the avalanche speed be derived from these timings? Can that be linked to snow conditions, size, path? That would tremendously increase the significance of the manuscript.

**We wanted to provide a concise overview of the course of events that we recorded, including the timings, also to be used for future comparison (with following events). Although we would have liked to include additional information from other sources (such as snow conditions, size, path, speed, …), we lack that information, unfortunately. Webcam footage (provided externally) is only one image per hour, and it only covers the road and part of the cones, not the whole mountain face (and it is not always visible due to snow/ice covering the lens). The webcam images were used to check the presence of avalanche events within the previous hour (through the presence of avalanche cones) as well as to get information about road traffic to be used for signal comparison.**

We do not have direct access to infrasound recordings and are only able to get that information from a webpage www.regobs.no (time, duration and likely location of the event). The test site lies in a remote mountain area, hence potential manual entries to the webpage from observers are also sparse.

**Line-specific Comments/Questions:**

**Abstract:**

L7: '… avalanches approach towards the fiber.'

Fixed.

**1 Introduction:**

L48: Typo: kHz instead of khZ

Fixed.

L59-61: Repetitive. I would recommend shortening or remove entirely.

Removed.

**2 Monitoring Site Description**

L75ff: I get a bit lost reading all the slope aspects and must go forward and backward to Fig. 1 and within the paragraph. For me it would help labeling the slopes A,B,C or similar, if this is applicable. However, it looks like the paper only deals with avalanches from the easternmost slope. If so, is there a necessity to indicate the other slopes at all?

Both Figures as well as the description were adapted and now only describe the main slope on the east facing mountain side.

L92: Wordy. I would suggest: 'We recorded strain rate with an ANS OptoDAS interrogator with 2m channel spacing, variable gauge lengths between 3.1m and 5.1m and sampling rates of 250 and 500 Hz for different seasons.' Or something similar.

Adjusted accordingly.

**3 Signal Classification**

**3.1 Types of Signals**

L101: Something is wrong with the wording of this sentence. Also, what makes a signal 'important'? Are these 4 signals most prominent or clean or representative?

We presented representative examples of the four signal types.

L102ff: The timing information is missing. It would be great for reproducibility if the exact timing information in UTC is provided.

Precise timing information is added (i.e. this event is 2022-02-10 09:00:59 UTC).

120: So why is the explosive signal not visible in Fig. 3a? What is the timing between the explosive signal and the avalanche signal. Are different frequency filters applied to a) and d)? - Note: This becomes clear later in the manuscript. However, it may be worth mentioning here.

We added a short explanation.

L125: Some wording seems odd here.

We shortened the sentences to make it more concise.

L134/135: I don't understand that reasoning. A broad frequency content comes from a short signal. That fits well into your theory of an impulsive impact at the cones.

**The reasoning was that the avalanche was large enough to reach the cones, and due to the impact, the signal is prominent. This is now corrected.**

**3.2 Signal evolution**

L148: What is being stacked for the PSDs? The waveforms before calculating the PSDs? Or the PSDs of individual channels? In this case, the latter would be the appropriate way to avoid negative interference. Also, what you call PSDs, I would call 'spectrograms'. Of course, each spectrogram time segment represents the PSD.

**We stacked the PSDs of the individual channels. We added a sentence to make this clear.**

L165: 'Due to the lack of additional data regagding the event, the reason for this extended duration remains unknown.' To me it's obvious that either one avalanche propagated faster than the other, or that the propagated distance to the cones differs. Is that a wrong or unreasonable assumption?

**This is reasonable.**

L168ff: This is important and should be described more concise. I start to understand only at this point, what you measure, as here you describe that you sense the seismic wave arrivals. It is written a bit unclear, such that I thought the 'seismic noise' comes from the impact on the cones, which does not make sense, because of the timing. Make clear that the 'seismic noise' corresponds to the avalanche propagation before the avalanche front hits the cones. Also, I think you should avoid the term 'seismic noise'. It is a clear signal. Probably also with coherent arrivals along DAS channels. I would assume that this signal is comprised from Rayleigh waves.

**We will be more concise in the revised manuscript.**

**4 Discussion**

L175: Where is this shown? You don't show plots where it gets clear if the road is reached or not it is not clear to me what signal content corresponds to what size of avalanche and how it varies significantly.

**This is a valid point; we removed the second part of the sentence. As mentioned earlier, we are dealing with lots of uncertainties regarding the events (size, speed, type of avalanche, path, amount of snow), this is now explained in the revised manuscript.**

L180: This sounds correct to me. It's much clearer than what is written in L168ff.

L192: Remove 'slightly'.

**Removed.**

L193: 'Although this is not verified yet, we aim to investigate this further using avalanche simulations.' Where? If you don't present it, there is no point in writing this.

**We recently performed (simple) avalanche simulations using RAMMS:avalanche, which show 2 snow accumulation areas near/at the cones: at the northern part of the cones and at the southern end (see Appendix). However, this is an initial attempt, and we recognize its limitation (extent of the release area, snow height, friction parameter, …). Therefore, we removed the part in the revised manuscript.**

L216: 'We have made valuable observations…' I would suggest letting the reader decide if the observations are valuable.

**Will be removed.**

L224: 'across the entire frequency spectrum' This should be more specific.

**Figures:**

General: The unit on any axis representing strain rate should be s-1, not $d\epsilon/dt$, as $d\epsilon/dt$ is not a unit.

**While not being a base SI unit, Epsilon is still accepted when describing strain (i.e. nε for 1e-9). Still, we adjusted this in all relevant Figures.**

Fig. 1: I got confused by the zoomed inset map (rightmost). I'd recommend a map that also includes roads, in particular the road that represents your fiber route. I thought your study site is between Geiranger and Erdal. The star in that zoomed inset is south-east of Jostedalsbreen.

**Good point, the position of the star is corrected and additional geographical context added (revised map is attached).**

Fig. 2: I personally don't like having a 2d birds-eye view and a 3d view of the study region in the same manuscript, because it's repetitive. I'd say in this case it's ok. However, I personally would put a figure like this in the supplementary information, as it is not essential for the manuscript.

**The Birds-Eye was moved to the appendix (and corrected to show only the eastern mountain face).**

Fig. 3: I strongly recommend using ISO 8601 as an international standard of displaying date and time. In your notation implies the American format and will certainly lead to misunderstandings.

**We now display date and time according to ISO 8601 in the updated manuscript.**

Fig. 4: I start wondering why you always show the 40s-time window for the explosion. I understand that you do it for consistency in Fig. 3. But this figure is really about the waveforms, correct? If the figure is only about the power per frequency interval, it would be much nicer to show a 'spatial spectrogram' with spectral power plotted versus DAS channels.

**We wanted to keep the time axis consistent for an easier comparison for the reader. We added F-X spectra in the appendix (see attachments). They highlight the difference in noise level across the array.**

Fig. 5: It is unclear to me, if the for the stacked PSDs the DAS channel waveforms were stacked, or if the PSDs of individual DAS channels were stacked. The upper row (a,b) has the time on the y-axis, whereas the lower row (c,d) has time on the x-axis. To me that's inconsistent, but not wrong. So it is your choice, but for future publications I recommend being consistent. For me, it helps understanding such multi-panel figures.

**For the two lower panels, the PSD's were computed for each channel individually and then stacked in the ranges that are highlighted in (a). We added a clarification about how (c) and (d) were computed.**

Figure label: A new date format again. I recommend using ISO 8601.

**For the revised manuscript we use the suggested date format as mentioned previously.**

Fig. 6: Similar to Fig. 5

Figure A1: a) I see 4 diagonal lines. Therefore 4 vehicles, not 6. Am I overseeing something?

**You are correct; this was an error and is now corrected.**

Figure A2: This is a great figure. We learn so much from it. There are coherent phase arrivals, you can fit a seismic wave speed (presumably a Rayleigh wave). As no information about the modeling is provided, I do not know what the amplitudes refer to. They also do not have a unit. In my opinion it is already enough to only have a constant velocity-travel time curve, i.e. a curved line representing the first arrival. Due to the steep topography, I think you should do it in 3D. This should result a longer distance traveled and therefore higher wave speed. I would expect a value between 2500-3300 m/s for Rayleigh waves.

**Adjusted to a single travel time curve (attached). The seismic velocity for the direct wave model was set to 2500 m/s, based on the rock type Gneiss (Orthogneiss in that area).**

**Tables:**

Table 1: I think this could go in the supplementary information as it is not essential to understand the manuscript.

**Table is moved to the Appendix.**

**Data availability.**

This seems useless. In my opinion it is ok if the data is not publicly available. I know that this does not agree with some journal policies. I face similar problems as you do with my DAS data. I would simply write: 'At the moment, the DAS data of this study not publicly available. Access can be granted on individual request.'

**Adjusted as recommended.**

**Attachements:**

**1. Revised Map**

Road was added, tunnel and gallery sections are displayed as dashed lines. The fiber section used for this study is displayed in magenta. For the overview map, context was added.

[Figure]

**2. Birds Eye View (moved to Appendix)**

[Figure]

**3. Traveltime curve**

**Adjusted as recommended with a single traveltime curve on top of the DAS section.**

[Figure]

**4. Trigger comparison**

The following images display three different trigger mechanisms – simple threshold, STA/LTA and Kurtosis. The rightmost column shows the summed triggers for all events (summed in blocks of 1 second each).

For each algorithm, three thresholds were tested (displayed in different rows).

**(a) Simple Threshold trigger**

[Figure]

**(b) STA/LTA**

[Figure]

**(c) Kurtosis**

[Figure]

**5. F-X Plots**

[Figure]

[Figure]

**F-X plot**
**explosion signal 2022-02-10 | highpass_0.5Hz**
**2022-02-10 09:00:19 UTC**

[Figure]

**6. Simulation (RAMMS:avalanche)**

---

## Author Comment (AC2)

**Reply to Martijn van der Ende**

Thank you for your comments and suggestions.

**1. General questions and comments**

How often would false detections be produced by road traffic?

We did not run a continuous trigger algorithm over the whole dataset. Consequently, we cannot quantify the trigger efficiency as of now. However, comparing the confirmed avalanche signals with road traffic examples (individual cars and trucks, see attached image for comparison), simplified detectors will likely end up with false detections due to traffic. The noise situation is also changing with the season, so an optimization process to find the best trigger values would be required. In this paper, we did not aim for a robust and automatic detector to only trigger on avalanches and not on cars, so we did not further tune the parameters to avoid traffic triggers.

Does it help to stack the detection metric over e.g. 100m of cable?

In principle, spatial stacking of data over several traces should increase the signal to noise ratios, yes. However, we also observe that the noise is spatially correlated, sometimes over many traces as well. Below we show some of our trials on stacking for three different triggering algorithms (raw data, STA/LTA, Kurtosis), see Appendix 1. Stacking does not solve the problem, unfortunately. Although large events (like the avalanche from 10 April 2022) can be distinguished from road traffic, stacked detection metrics for smaller ones (like the one from 2 February 2024) lie in the same range as traffic.

Or apply a low-pass filter in space, given that the main avalanche signals have a spatial footprint that is larger than that of individual vehicles on the road?

The spatial footprint is not necessarily larger than that of individual vehicles – it can be similar when the avalanche event is not as large and more localized (see natural avalanche from 2 February 2024).

Or apply a basic FK-filter to remove the characteristic vehicle speeds?

We appreciate the constructive comments, and we will also investigate further in this direction of taking advantage of the high spatial resolution of the DAS data. We actually did apply and try quite a few different signal processing steps that unfortunately did not improve the results significantly above the STA/LTA approaches (semblance, semblance weighted, cross-correlation weighting, kurtosis/period,...). In the revised version we will include some of these approaches in the appendix.

Aside from this comment, I only have a few tiny suggestions for improvement:

- The authors use "spiky" to describe some of the more localised signals. Perhaps "impulsive" could be a more formal alternative description, but I leave this entirely up to the authors to consider.

  Adjusted as recommended.

- I think that the readability of Section 3.1 could improve by splitting the big paragraph (lines 101-138) into a few individual paragraphs.

  Adjusted as recommended.

- The reference to Xie et al. on line 210 is missing a date. Also, there are several papers that describe DAS-based vehicle detection that are already published, so I would suggest citing one or two of those in addition to this preprint.

  **Fixed and additional reference was added.**

- A suggestion for future work, if the authors intend to remove traffic signals from their data/detections: the traffic signals in the 0.5-5 Hz frequency band are very simple, as they basically represent the spatial footprint of a point load indenting a surface (which translates in time and space; see the Methods section of Jousset et al., 2018; https://www.nature.com/articles/s41467-018-04860-y). It is possible to deconvolve this characteristic signal from the data, which yields high-resolution detections of each vehicle on the road (https://dl.acm.org/doi/10.1109/TITS.2023.3322355; preprint: https://arxiv.org/abs/2212.03936). The authors could use this approach to exclude STA/LTA detections associated with traffic, or to mask portions of the data that coincide with traffic signals. This is merely a personal suggestion for future work, I don't expect the authors to expand on this for the present study.

  **Thank you for the recommendation, this looks very interesting.**

Kind regards,

Martijn van den Ende

**Attachements**

**1. Trigger comparison**

**The following images display three different trigger mechanisms – simple threshold, STA/LTA and Kurtosis. The rightmost column shows the summed triggers for all events (summed in blocks of 1 s each). The triggered avalanche is**

**For each algorithm, three thresholds were tested (displayed in different rows).**

**(a) Simple Threshold trigger**

[Figure]

**(b) STA/LTA**

**(c) Kurtosis**

[Figure]

**2. F-X Plots**

[Figure]

[Figure]

F-X plot
car 2022-02-10 | highpass_0.5Hz
2022-02-22 11:07:27 UTC

[Figure]

F-X plot
explosion signal 2022-02-10 | highpass_0.5Hz
2022-02-10 09:00:19 UTC

---

## Author Comment (AC3)

**Answer to David Walter**

**Thank you for your detailed comments.**

Kleine et al. describe a seismic data set of snow avalanches. The novelty is that the data were acquired with distributed acoustic sensing (DAS) using a pre-installed communication fiber running along an avalanche-prone road. In this regard, the presented work is highly relevant, because existing fiber optic infrastructure provides an unrivalled density and coverage of sensors for the detection of and possibly warning against avalanches and other hazardous mass movements. The manuscript is written clearly and the data are of good quality and benchmarked against independent acoustic records. On the other hand, the study makes use of only a small part of the available information: 1. It is not clear how many other avalanches were recorded that are included in the Vasom Regobs catalogue. 2. The signal analysis could be done in more depth. 3. Additional ground observations such as avalanche sizes and locations identified on the webcam or infrasound data could be taken into account for the analysis (based on the manuscript, these data sources exist). For a typical NHESS paper I would expect the authors to take at least one of the above steps beyond the current analysis state. For this reason, I recommend that this paper be rejected in its current form. On the other hand, expanding the paper along one or several of the suggested lines should be straightforward requiring mostly a deeper dive into what the authors have at hand and already partially done. From my point of view, it is not necessary to apply advanced data processing, location algorithms and signal classification, although this should also not be difficult to do. Alternatively, the authors may consider submitting the manuscript as a brief communication. For this, some consolidation of text and figures as well as the addition of more specific information would suffice.

Fabian Walter.

SPECIFIC COMMENTS

In its current state, the manuscript seems to be mainly a first glance at the data. To expand it towards a full-size paper it requires an exhaustive investigation in one of the above-mentioned directions. Here are some suggestions for further data analysis and discussion:

1. Spatial coherence: Are individual phases visible over longer distances? I would expect this if collisions with the cones are indeed seismogenic. How do the signals look like before and after crossing the cable? Are similar moveouts as for the explosion signal also seen in the natural signals? Assuming Rayleigh waves (often dominating high-frequency mass movement signals), different parts of the cable should show a different sensitivity to the approaching avalanche. The closest cable sections should be least sensitive assuming a perpendicular approach unto a straight cable section (Kennett et al., 2024 in GJI). Can this be observed? Are there relevant low-frequency signals that can be attributed to the avalanches' weight? Can the signals be used to differentiate between avalanche paths (see specific comment below)?

   **Thank you for this comment. Spatial coherence is indeed expected and should be exploited due to the high spatial resolution in DAS data. However, investigation of several spatial signal processing approaches like semblance-weighted stacks, spatial cross-correlation and FK-filters have not provided significant improvement for the classification of wavetypes. The geometry of the cable towards the protective cones does not allow to investigate changes in the wavefield before and after the avalanche hits the cones. That**

**would indeed be of high interest to investigate and plans to extend the cable up along the hillside already exist but could not be yet realized due to practical issues.**

2. Discussion: The first discussion paragraph suggests that the authors sorted their avalanches by size. How was this size information obtained? Currently, the reader cannot verify which avalanche records correspond to which size of events. A systematic comparison between size, signal strength, central frequency or other waveform characteristics would be interesting.

**We only have a small number of verified avalanche events to work with and only have limited information about those. That is why a systematic comparison is not possible for this dataset. The size information that was included here stems from the webpage www.regobs.no entries.**

**Webcam footage (provided externally) is one image per hour, and it only covers the road and part of the cones, not the whole mountain face (and it is not always visible due to snow/ice covering the lens). The webcam images were used to check the presence of avalanche events within the previous hour (through the presence of avalanche cones) as well as to get information about road traffic to be used for signal comparison. We do not have direct access to infrasound recordings and are only able to get that information from Regobs (time, duration and likely location of the event). The test site lies in a remote mountain area, hence manual entries from observers are also sparse.**

I was not able to picture the avalanche/road/cable setting. I strongly suggest that the authors include one or more valley cross-sections, even if this is only showing a schematic not to scale. This would answer questions on how an avalanche reaches the road, the cable, the topographic step next to the parking lot or other sites.

**We simplified the maps and now only show the eastern mountain face.**

MISCELLANEOUS COMMENTS

Avalanche events --> avalanches

**Fixed.**

Lines 21-22: Delete sentence starting with "Hence, …" (trivial content)

**Fixed.**

Line 28: delete "reliable", since more is needed for a reliable tool than just independence of meteorological conditions

**Fixed.**

Line 31: specify characteristics; which "same test site"?

**We added the information, it is the Vallée de la Sionne avalanche test site.**

Lines 32-33: approaching which geophone? The reader needs more information, like avalanche size, source-station distance, …

**We will add this information.**

Line 37: localizing --> locating

**Corrected.**

Lines 46-47: delete paragraph break

**Corrected.**

Line 68: rewrite "beginning and end"

**Corrected.**

Lines 72-73: Indicate infrasound system on Figure 1 or 2.

**Done.**

Lines 75ff: The described avalanche slopes should be labeled (or color-coded) on Figure 1.

**Map was simplified, now only the eastern facing slope is shown.**

Line 84: steepness --> slope

**Corrected.**

Lines 91-92: remove "below in the"

**Corrected.**

Line 92: remove "during data handling"

**Corrected.**

Section 3.1: Did you also record earthquakes?

**We recorded at least one regional earthquake, which was seen over the whole stretch of the cable. We though did not further analyze the event.**

Line 98: Check Varsom Regobs, there are two entries in the bibliography.

**One entry is for the website as such, the other one for a specific event.**

Line 99: Explain the infrasound measurements. Is this from antenna data?

**We do not have direct access to infrasound measurements, we only access the automatic detections on Varsom RegObs.**

Line 101ff: Paragraph is too long. Some numbers are spelled out, others are not. When describing channel sections and geographic directions, it would help to label channel distances in one of the plots, or at least refer to the color code in one of the figures.

**Good point, we will split the paragraph and refer to the color codes.**

Line 112: I would drop a hint here about the source of the four signals (even though they are discussed in more detail later).

**This will be added.**

Lines 124-125: The modeling and the "similar behavior" have to be explained. Especially the modeling requires details.

**We computed first arrival travel times using a simple shortest path method—assuming constant velocity and neglecting topography. We will add more details to clarify this calculation.**

Line 128-129: Where does the information on the avalanche location come from? Why is this not shown somewhere?

**The information is taken from Varsom RegObs, we will rephrase this sentence.**

Lines 134-135: The broadband character can have different reasons and influences. I would include more discussion here or leave out this reason.

**This was also mentioned in the first comment. We wanted to establish a link to the collision of the snow mass with the cones, resulting in a broadband signal. Since the sentence was ambiguous in the manuscript, we will make this point clear.**

Lines 137-138: I cannot verify the 16 s and the 20-100 Hz range. The figure seems to suggest a different burst signal in b) and a different frequency range.

**This sentence refers to the traffic signals, and the one at 16 s is visible on every frequency band. We will break this into a new paragraph to improve readability.**

Line 139: Avoid 1-sentence paragraphs. Also, what does "strong" refer to? A high signal-to-noise ratio?

**The most energy is found in this frequency range, and while large events are visible in all frequency ranges, smaller ones can only be identified in this band.**

Lines 150ff and 160ff: I could not confirm most of the exact time information.

**Thanks for pointing that out – we got the text for the figures mixed up. This will be corrected.**

Line 165: Can the different duration be related to different flow velocities?

**No, as we do not have additional information about the velocities.**

Lines 182ff: It would help to see more quantitative information from the Surinach studies (avalanche sizes, source-station distances, sensor types, …).

**We will add those in the revised manuscript.**

Lines 192: The DAS signals should provide further hints with respect to the different avalanche paths. This is the kind of additional analysis that I strongly suggest (see general comments above), even if this is done on a qualitative level.

**As mentioned, we do not have much meta data about the avalanches.**

Lines 207ff: I strongly suggest discussing Kang et al., 2024, in GRL, here.

**Thank you for the reference, we will look into this.**

Line 210: No year for reference.

**Fixed.**

Lines 204-205: Is it really this simple to distinguish car traffic? What about cars leaving from the parking lot or moving across it?

**This is an important point. Distinguishing car traffic is not straightforward, especially in complex scenarios like vehicles leaving the parking lot or moving across it. However, most of the traffic signals during the 3 seasons are visible as inclined lines on the DAS sections.**

Line 212: "Sobel operator" needs a reference.

**Will be added.**

Conclusion is a summary. I suggest deleting it and calling the Discussion section "Discussion and Outlook".

**Thanks for the suggestion, we will adjust this accordingly.**

FIGURES AND TABLE

Captions should not include paragraph breaks.

**Good point, this was fixed.**

Table 1: The meaning of the gauge length is not explained. Either include it or remove the information from the table.

**We would argue that gauge length is a well-known term in the DAS community and does not need further description.**

Figure 1 caption: I suggest rewriting "discriminability".

**The whole Figure was changed and simplified.**

Figure 2: I suggest labeling the cones, also in the inset; What is Saetreskarsfjellet? Is this mentioned/defined in the text somewhere? Caption: correct typo(s); "start of" ◊ "entrance of".

**We added an explanation (Saetreskarsfjellet is the mountain where the avalanche prone slope is located).**

Figure 3: Panel b: It seems that the avalanche is visible on practically all channels. I suggest pointing this out or even amplifying the signal to make this more obvious. Panel d: I suggest decreasing the extent of the y-axis so that the explosion moveout can be seen here.

**The signal that is visible on all channels in panel b is due to noise (this is a problem we encountered often in the 2024 dataset).**

Figure 4: I suggest labeling the columns (rather than defining them in the caption) in their top panels (using titles or text boxes).

**This will be adjusted accordingly in the revised manuscript.**

Figure 5: Here and perhaps in other figures, I suggest increasing the font sizes of some of the text. The precursory signal in b) (circled in white dashed) is not visible.

**Thanks for the suggestions, we will improve readability.**

Figure 6: The caption seems to be equivalent to Figure 5, so I suggest replacing it by a respective sentence.

**This was adjusted accordingly.**

Figure 3 and elsewhere: the colors of the waveforms are difficult to discern, I am not sure the color scale serves its purpose.

**Attachements:**

**Revised Map**

Road and infrasound were added, tunnel and gallery sections are displayed as dashed lines. The fiber section used for this study is displayed in magenta. For the overview map, context was added.